## [Peer Review File · Nature Communications]

Reviewers' comments:

Reviewer #1 (Remarks to the Author):

The authors have carried out a difficult sampling and analytical task (the analysis of Pb and Pb isotopes in seawater) and produced excellent data that is a significant contribution to the understanding of pollutant and natural Pb in the ocean geochemical system. The paper is well written and very well thought out, and I have only a few comments that should be considered in the final version:

(1) I think the authors should consider the possibility that the low 6/7 values in the W. tropical Atlantic could be due to transport of Brazilian lead from the southern hemisphere. I have data for surface waters between the equator and Rio de Janeiro and it is consistent at ~ 1.165 , similar to the polluted sediments in Guanabara Bay (M.C. Geraldès et al., 2006, *Journal of Geochemical Exploration* 88: 384-388). The attached figure shows that the western N. Atlantic receives a strong flow from south of the equator.

(2) I think the authors are stretching with their interpretation that the Pb residence time is a few months. They haven't demonstrated any seasonal variability, and in fact they admit that their data could be explained by the seasonal dominance of the dust fluxes to the two regions.

Reviewer #2 (Remarks to the Author):

This manuscript presents Pb concentrations and isotope ratios in the surface waters and aerosols in the tropical North Atlantic Ocean, and reports the first observation of significant proportions of natural Pb in those samples as a result of global phase-out of leaded gasoline. It constitutes a unique set of data with great quality. I find it particularly impressive that they measured elemental ratios and Pb isotopes from acid-leached and total-digested fractions of the aerosols, separating out anthropogenic Pb isotopic signals effectively from naturally-occurring Pb, and also providing insights on the importance of solubilities and deposition rates on the supply of different types of Pb to the surface ocean. This is a well written paper with a discussion based on solid data. I strongly recommend this manuscript to be published in the *Nature communications* as the results of this study should be interesting to a wide audience, but I think some issues need to be addressed to strengthen their conclusions.

- The authors explain the Pb in the surface waters as a result of atmospheric Pb inputs, but they need to add a discussion on the effect of advection on the distribution of Pb isotopes in the surface waters. They can also include in the introduction a short description on surface ocean currents in the studied area.

-This study was conducted in the tropical North Atlantic Ocean, where dust inputs are the largest in the world ocean, but anthropogenic Pb emissions from nearby countries are historically lower. Detection of high % of natural Pb may be confined to this area, and other parts of the Atlantic Ocean, e.g., subtropical, may still be dominated by anthropogenic Pb. Moreover, from Fig. 3 in this manuscript (surface waters from 1980s and 1990s), it seems that even in old days when anthropogenic Pb emissions were greater than now, naturally occurring Pb still consisted 10-15% of the surface ocean Pb in this region. In this sense, the title of this manuscript - "return" of naturally sourced Pb to "Atlantic" surface waters - seems to be somewhat misleading.

Line 136-146. Choosing end-members is very important in quantifying inputs from different Pb sources, especially because these end-member values change with time. The authors made a careful approach to determine valid end-members for each source, but the aerosol data they used to determine Pb isotope ratios for North American Pb sources are from two decades ago. Recent North American Pb sources may have different isotope ratios from those times. For example, Noble et al. (2015, Deep-Sea Research II, 116, pp. 208-225) found the modern North American aerosols have lower 206/207 ratios relative to historical values. Their values fall closely to the mixing line between North African anthropogenic Pb and mineral dust Pb in Fig. 3 of this manuscript, so, in fact, North American anthropogenic Pb sources may not be clearly distinguishable from the other sources when this study was conducted.

Line 210-225. Please check if the assumption of binary mixing is still valid with updated North American Pb isotope ratios. I guess it would still be okay to assume African anthropogenic and African dust Pb sources are the major ones given the wind directions dominating over this region, but some samples collected from higher latitudes may have been affected by North American Pb sources as well.

Line 221. It would be helpful including the %Pb_{min} contribution for each sample in Table 1.

Line 295. Provide an error range for this Pb_{min} estimates. Maybe take some error range for the Pb/Al_{ucc} value and propagate over the calculation, like the authors did in line 269?

Line 301. 25% difference?

Line 359. The result of this study clearly shows the effect of the phase-out of leaded gasoline, but as the authors point out later, anthropogenic Pb emissions from other sources are still significant, and non-gasoline Pb emissions are actually increasing in some other parts of the world (e.g., coal burning from China).

Line 419-426. Atmospheric Pb_{min} concentrations are based on mineral dust concentrations in the WTA, whereas atmospheric Pb_{anthr} concentrations are from the aerosols collected in the ETA. Atmospheric Pb_{anthr} concentrations in the WTA should be smaller than those in the ETA due to a longer distance from the source regions. For instance, assuming the atmospheric Pb_{anthr} concentrations in the WTA is 2-4 times lower than ETA (from Figure 1), a reasonable range for the (Pb_{min}/Pb_{anthr})_{atm} ratio seems to be around 4.

Reviewer #3 (Remarks to the Author):

Review of "Return of naturally sourced Pb to Atlantic surface" by Bridgestock et al.

Comments:

This paper presents a major effort to quantify the sources of dissolved Pb (natural vis-à-vis anthropogenic) to the tropical Atlantic Ocean by measuring Pb concentration and their isotopic composition in surface seawater samples as well as atmospheric aerosols collected during two GEOTRACES cruises in western and eastern tropical North Atlantic. The two cruises were strategically conducted during different seasons to transect the North African dust plume in order to assess the role of transported African dust as a significant contributor of dissolved Pb to the surface water of tropical Atlantic. This paper claims significant proportions of up to 30 - 50% of natural Pb, derived from mineral dust, are observed in Atlantic surface waters, post global phasing out of leaded gasoline. Further, based on the Pb isotope ratios and concentration, they find that dry deposition of atmospheric mineral dust is the dominant process which governs the dissolved Pb concentration of surface seawater of tropical Atlantic. Both of their findings are well supported and corroborated with the present data set.

One of the interesting parts of this paper is that they have used two different approaches (1) isotope mass balance calculation and (2) crustal enrichment factor calculation to estimate atmospheric Pb from mineral dust and/or anthropogenic sources (Pb_{min} and Pb_{anthr}). Both approaches lead to concurrent findings of high variability in the relative proportion of Pb_{min} and Pb_{anthr} in atmospheric aerosols over

tropical Atlantic and a significant contribution of up to 30 - 50% of Pb are derived from natural source (mineral dust) to the Atlantic surface water. Further, they have successfully demonstrated using a sensitivity test that dry deposition is dominant key factor in producing the high proportion of Pb_{min} in the Atlantic surface water. Although, these findings are limited to regional latitudinal belts which are mostly influenced by North African dust plumes.

The manuscript is well written, easy to read and should be of great interest to the large community of Ocean and atmospheric sciences. This paper provide concrete evidences of the role of atmospheric mineral dust in modulating the dissolved Pb concentration in Atlantic surface waters (although regional in nature) and advances our understanding in the biogeochemical cycling of Pb. I thoroughly enjoyed reading this well drafted paper, with no errors and, thus, strongly recommend for the publication in its present form.

Bridgestock et al., Return of naturally sourced Pb to Atlantic surface waters

Responses to the referee's comments

Reviewer 1;

(1) I think the authors should consider the possibility that the low 6/7 values in the W. tropical Atlantic could be due to transport of Brazilian lead from the southern hemisphere. I have data for surface waters between the equator and Rio de Janeiro and it is consistent at ~1.165, similar to the polluted sediments in Guanabara Bay (M.C. Geraldles et al., 2006, Journal of Geochemical Exploration 88: 384-388). The attached figure shows that the western N. Atlantic receives a strong flow from south of the equator.

- We appreciate this comment and acknowledge that northward advection of Brazilian Pb from the southern hemisphere could have affected the seawater composition in the western tropical Atlantic (WTA). We added a sentence to this effect at lines 147-149.

(2) I think the authors are stretching with their interpretation that the Pb residence time is a few months. They haven't demonstrated any seasonal variability, and in fact they admit that their data could be explained by the seasonal dominance of the dust fluxes to the two regions.

- Although tentative, it is an important observation that maximum Pb_{min} contributions appear to be recording the position of seasonally dynamic atmospheric inputs. Further arguments supporting the assertion of the short residence time of these input signals have been developed by taking into account the expected effects of circulation on the distribution of Pb (lines 364-368). The language in this paragraph has also been altered to reflect the more speculative nature of this particular suggestion.

Reviewer 2;

(1) The authors explain the Pb in the surface waters as a result of atmospheric Pb inputs, but they need to add a discussion on the effect of advection on the distribution of Pb isotopes in the surface waters. They can also include in the introduction a short description on surface ocean currents in the studied area.

- Surface circulation can act to redistribute and mix the different atmospheric Pb inputs to surface waters, and this is now acknowledged in lines 364-368. We also added a comment about the significance of northward advection in the WTA (lines 147-149; see comment above). In our opinion, a full description of surface ocean currents in the tropical Atlantic would only be warranted, if the seawater Pb signals were inconsistent with the main atmospheric inputs from easterlies and westerlies (i.e. the main suppliers of Pb to the surface ocean). The fact that our surface water Pb data carry information on the (seasonal) position of mineral dust deposition hence corroborates our approach to neglect a more detailed discussion of surface ocean circulation in the area.

(2) This study was conducted in the tropical North Atlantic Ocean, where dust inputs are the largest in the world ocean, but anthropogenic Pb emissions from nearby countries are historically lower. Detection of high % of natural Pb may be confined to this area, and other parts of the Atlantic Ocean, e.g., subtropical, may still be dominated by anthropogenic Pb. Moreover, from Fig. 3 in this manuscript (surface waters from 1980s and 1990s), it seems that even in old days when anthropogenic Pb emissions were greater than now, naturally occurring Pb still consisted 10-15% of the surface ocean Pb in this region. In this sense, the title of this manuscript - "return" of naturally sourced Pb to "Atlantic" surface waters - seems to be somewhat misleading.

- The historical surface water data do not show a clear deviation towards the composition of mineral dust in Fig. 3, lying within/close to the defined anthropogenic fields. Due to the uncertainty of the true average compositions of the different anthropogenic endmembers, it cannot be confidently stated that these historical samples contained significant (10-15%) proportions of Pb_{min} . Furthermore, Pb concentrations in surface waters of this region were ~10 times higher in 1989 than 2005, indicating that these waters were once overwhelmed by anthropogenic inputs (Pohl et al., 2011, Journal of Marine Systems, 84, pp 28-41). Hence, we maintain that our study is the first to definitively identify significant proportions of natural Pb in surface waters of the Atlantic, and that surface waters of the tropical Atlantic were dominated by anthropogenic Pb sources during the past few decades. We hence maintain that the chosen title of the manuscript is appropriate. We have, however, changed the wording in line 87 to acknowledge that small amounts of natural Pb may well have been present in the tropical North Atlantic even at the height of Pb inputs from leaded gasoline.

(3) Line 136-146. Choosing end-members is very important in quantifying inputs from different Pb sources, especially because these end-member values change with time. The authors made a careful approach to determine valid end-members for each source, but the aerosol data they used to determine Pb isotope ratios for North American Pb sources are from two decades ago. Recent North American Pb sources may have different isotope ratios from those times. For example, Noble et al. (2015, Deep-Sea Research II, 116, pp. 208-225) found the modern North American aerosols have lower 206/207 ratios relative to historical values. Their values fall closely to the mixing line between North African anthropogenic Pb and mineral dust Pb in Fig. 3 of this manuscript, so, in fact, North American anthropogenic Pb sources may not be clearly distinguishable from the other sources when this study was conducted.

Line 210-225. Please check if the assumption of binary mixing is still valid with updated North American Pb isotope ratios. I guess it would still be okay to assume African anthropogenic and African dust Pb sources are the major ones given the wind directions dominating over this region, but some samples collected from higher latitudes may have been affected by North American Pb sources as well.

- The recent North American aerosol data of Noble et al. (2015) are now included in the source assessment, and are denoted in Fig. 3 with their own symbols. These data lie at the lower end of the previously defined North/Central American anthropogenic emission field, and are still

distinguishable from North African emissions. Importantly, the surface water samples from north of 20°N in the WTA evolve towards the composition of these more recent aerosol data, consistent with our interpretation that these waters contain increased contributions from North American emissions (lines 155-158). Regarding the second point, we can confirm that the assumption of binary mixing for the samples featuring the highest Pb_{min} contributions is still valid.

(4) Line 221. It would helpful including the %Pb_{min} contribution for each sample in Table 1.

- %Pb min contributions for individual samples were (and are) intentionally omitted, as they would imply a precision in our estimates (i.e. at the percent level), which is not justified. In detail, there is uncertainty in how well the averages of the compiled literature datasets match the true composition of the endmembers. Hence a conservative range of 30-50% is quoted for the samples with the highest Pb_{min} contributions.

(5) Line 295. Provide an error range for this Pb_{min} estimates. Maybe take some error range for the Pb/Al_{ucc} value and propagate over the calculation, like the authors did in line 269?

- The results from the isotope mass balance approach are used to guide the choice of the Pb/Al ucc ratio, with that of Rudnick & Gao (2003) providing the best fit. Since it can be inferred from this agreement that the chosen Pb/Al ucc ratio is a good approximation for the composition of the mineral dust in our samples, varying this ratio (and hence creating poorer agreements between the two approaches) seems an inappropriate way to assess uncertainty. In fact, it is the confidence in the chosen Pb/Al ucc ratio that allows us to use our enrichment factors in a more quantitative manor than in the conventional approach (see lines 276-279), which suffers from uncertainty in the appropriate Pb/Al ucc ratio. We therefore suggest that the best way to assess uncertainty of the Pb_{min} % estimates using the EF approach is by direct comparison to the independent estimates derived from the isotope mass balance approach.

(6) Line 301. 25% difference?

- Has been corrected to 25%.

(7) Line 359. The result of this study clearly shows the effect of the phase-out of leaded gasoline, but as the authors point out later, anthropogenic Pb emissions from other sources are still significant, and non-gasoline Pb emissions are actually increasing in some other parts of the world (e.g., coal burning from China).

- We agree with this statement and reworded the sentence to refer specifically to circum-Atlantic regions (line 353-354).

(8) Line 419-426. Atmospheric Pb_{min} concentrations are based on mineral dust concentrations in the WTA, whereas atmospheric Pb_{anth.} concentrations are from

the aerosols collected in the ETA. Atmospheric Pb_{anth} concentrations in the WTA should be smaller than those in the ETA due to a longer distance from the source regions. For instance, assuming the atmospheric Pb_{anth} concentrations in the WTA is 2-4 times lower than ETA (from Figure 1), a reasonable range for the (Pb_{min}/Pb_{anth})_{atm} ratio seems to be around 4.

- Constraints on monthly average mineral dust concentrations in the ETA have been incorporated using time-series data from the Cape Verde Islands (Patey et al., 2015, Marine Chemistry, 174, pp 103-119). A geometric mean of the atmospheric Al concentration of 2.4 $\mu\text{g m}^{-3}$ is given for the seasonal dusty period (January-February, 2008) in this region, corresponding to a mineral dust concentration of about 25 $\mu\text{g m}^{-3}$. This is within the range of the monthly averaged mineral dust concentrations from sites in the WTA previously used (20-50 $\mu\text{g m}^{-3}$; Prospero et al., 2014, Global Biogeochemical Cycles, 29, pp 757-773). The text in lines 413 to 414 has been clarified accordingly.

REVIEWERS' COMMENTS:

Reviewer #1 (Remarks to the Author):

I am satisfied that the revisions meet my comments and those of the other reviewers. Although I still disagree on the interpretation of the short Pb residence time, it is now sufficiently qualified and labeled speculative, so I believe that the readers are appropriately warned that this is not a hard and fast conclusion.

Reviewer #2 (Remarks to the Author):

All major points were sufficiently addressed in the revised manuscript and the author's response to the comments. Added discussion on the residence time of Pb in the surface waters in the studied region now clarifies their point and makes this manuscript even more interesting. I do not see other errors in the revised manuscript, and therefore, strongly recommend for the publication.

Response to the referee's comments

Reviewer #1 (Remarks to the Author):

I am satisfied that the revisions meet my comments and those of the other reviewers. Although I still disagree on the interpretation of the short Pb residence time, it is now sufficiently qualified and labeled speculative, so I believe that the readers are appropriately warned that this is not a hard and fast conclusion.

Reviewer #2 (Remarks to the Author):

All major points were sufficiently addressed in the revised manuscript and the author's response to the comments. Added discussion on the residence time of Pb in the surface waters in the studied region now clarifies their point and makes this manuscript even more interesting. I do not see other errors in the revised manuscript, and therefore, strongly recommend for the publication.

-We appreciate the feedback the reviewers have provided on the manuscript and thank them for their efforts in contributing to its improvement.